# Snap ML: A Hierarchical Framework for Machine Learning

**Celestine Dünner**[*1]  **Thomas Parnell**[*1]
**Dimitrios Sarigiannis**[1]  **Nikolas Ioannou**[1]  **Andreea Anghel**[1]  **Gummadi Ravi**[2]
**Madhusudanan Kandasamy**[2]  **Haralampos Pozidis**[1]
[1]IBM Research, Zurich, Switzerland
[2]IBM Systems, Bangalore, India
{cdu,tpa,rig,nio,aan}@zurich.ibm.com
{ravigumm,madhusudanan}@in.ibm.com
hap@zurich.ibm.com

## Abstract

We describe a new software framework for fast training of generalized linear models. The framework, named Snap Machine Learning (Snap ML), combines recent advances in machine learning systems and algorithms in a nested manner to reflect the hierarchical architecture of modern computing systems. We prove theoretically that such a hierarchical system can accelerate training in distributed environments where intra-node communication is cheaper than inter-node communication. Additionally, we provide a review of the implementation of Snap ML in terms of GPU acceleration, pipelining, communication patterns and software architecture, highlighting aspects that were critical for achieving high performance. We evaluate the performance of Snap ML in both single-node and multi-node environments, quantifying the benefit of the hierarchical scheme and the data streaming functionality, and comparing with other widely-used machine learning software frameworks. Finally, we present a logistic regression benchmark on the Criteo Terabyte Click Logs dataset and show that Snap ML achieves the same test loss an order of magnitude faster than any of the previously reported results, including those obtained using TensorFlow and scikit-learn.

## 1   Introduction

The widespread adoption of machine learning and artificial intelligence has been, in part, driven by the ever-increasing availability of data. Large datasets can enable training of more expressive models, thus leading to higher quality insights. However, when the size of such datasets grows to billions of training examples and/or features, the training of even relatively simple models becomes prohibitively time consuming. Training can also become a bottleneck in real-time or close-to-real-time applications, in which one's ability to react to events as they happen and adapt models accordingly can be critical even when the data itself is relatively small.

A growing number of small and medium enterprises rely on machine learning as part of their everyday business. Such companies often lack the on-premises infrastructure required to perform the compute-intensive workloads that are characteristic of the field. As a result, they may turn to cloud providers in order to gain access to such resources. Since cloud resources are typically billed by the hour, the time required to train machine learning models is directly related to outgoing costs. For such an enterprise cloud user, the ability to train faster can have an immediate effect on their profit margin.

---

[*]equal contribution.

The above examples illustrate the demand for fast, scalable, and resource-savvy machine learning frameworks. Today there is an abundance of general-purpose environments, offering a broad class of functions for machine learning model training, inference, and data manipulation. In the following we will list some of the most prominent and broadly-used ones along with certain advantages and limitations.

*scikit-learn* [16] is an open-source module for machine learning in Python. It is widely used due to its user-friendly interface, comprehensive documentation and the wide range of functionality that it offers. While scikit-learn does not natively provide GPU support, it can call lower-level native C++ libraries such as LIBLINEAR to achieve high-performance. A key limitation of scikit-learn is that it does not scale to datasets that do not fit into the memory of a single machine.

*Apache MLlib* [10] is Apache Spark's scalable machine learning library. It provides distributed training of a variety of machine learning models and provides easy-to-use APIs in Java, Scala and Python. It does not natively support GPU acceleration, and while it can leverage underlying native libraries such as BLAS, it tends to exhibit slower performance relative to the same distributed algorithms implemented natively in C++ using high-performance computing frameworks such as MPI [6].

*TensorFlow* [1] is an open-source software library for numerical computation using data flow graphs. While TensorFlow can be used to implement algorithms at a lower-level as a series of mathematical operations, it also provides a number of high-level APIs that can be used to train generalized linear models (and deep neural networks) without needing to implement them oneself. It transparently supports GPU acceleration, out-of-core operation, multi-threading and can scale across multiple nodes. When it comes to training of large-scale linear models, a downside of TensorFlow is the relatively limited support for sparse data structures, which are frequently important in such applications.

In this work we will describe a new software framework for training generalized linear models (GLMs) that is designed to offer effective GPU-accelerated training in both single-node and multi-node environments. In mathematical terms, the problems of interest can be expressed as the following convex optimization problem:

$$\min_{\boldsymbol{\alpha}} f(A\boldsymbol{\alpha}) + \sum_i g_i(\alpha_i) \tag{1}$$

where $\boldsymbol{\alpha}$ denotes the model to be learnt from the training data matrix $A$ and $f, g_i$ are convex functions specifying the loss and regularization term. This general setup covers many primal and dual formulations of widely applied machine learning models such as logistic regression, support vector machines and sparse models such as lasso and elastic-net.

**Contributions.** The contributions of this work can be summarized as follows:

- We propose a hierarchical version of the CoCoA framework [18] for training GLMs in distributed, heterogeneous environments. We derive convergence rates for such a scheme which show that a hierarchical communication pattern can accelerate training in distributed environments where intra-node communication is cheaper that inter-node communication.

- We propose a novel pipeline for training on datasets that are too large to fit in GPU memory. The pipeline is designed to maximize the utilization of the CPU, GPU and interconnect resources when performing out-of-core stochastic coordinate descent.

- We review the implementation of the Snap ML framework, including its GPU-based local solver, streaming CUDA operations, communication patterns and software architecture. We highlight the aspects that are most critical in terms of performance, in the hope that some of these ideas may be applicable to other machine learning software, including popular deep learning frameworks.

## 2   System Overview

We start with a high-level, conceptual description of the Snap ML architecture. The core innovation of Snap ML is how multiple state-of-the-art algorithmic building blocks are nested to reflect the hierarchical structure of a distributed systems. Our framework, as illustrated in Figure 1, implements three hierarchical levels of data and compute parallelism in order to partition the workload among different nodes in a cluster, taking full advantage of accelerator units and exploiting multi-core parallelism on the individual compute units.

*Level 1*. The first level of parallelism spans across individual worker nodes in a cluster. The data is distributed across the worker nodes that communicate via a network interface. This data-parallel approach serves to increase the overall memory capacity of our system and enables the training of large-scale datasets that exceed the memory capacity of a single machine.

*Level 2*. On the individual worker nodes we can leverage one or multiple GPU accelerators by systematically splitting the workload between the host and the accelerator units. The different workloads are then executed in parallel, enabling full utilization of the available hardware resources on each worker node, thus achieving a second level of parallelism, across heterogeneous compute units.

*Level 3*. To efficiently execute the workloads assigned to the individual compute units we leverage the parallelism offered by their respective compute architecture. We use specially-designed solvers to take full advantage of the massively parallel architecture of modern GPUs and implement multi-threaded code for processing the workload on CPUs. This results in an additional, third level of parallelism across cores.

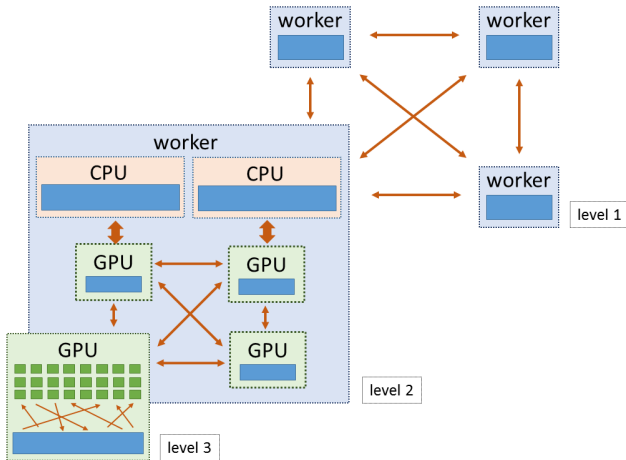

Figure 1: Hierarchical structure of our distributed framework.

## 2.1 Hierarchical Optimization Framework

The distributed algorithmic framework underlying Snap ML is a hierarchical version of the popular CoCoA method [18]. CoCoA is designed for communication-efficient distributed training of models of the form (1) across $K$ worker nodes in a data-parallel setting. It assumes that the data matrix $A$ is partitioned column-wise across the $K$ workers and defines independent data-local optimization tasks for each of them. These tasks only require access to the local partition of the training data and a shared vector $\mathbf{v}$. This shared vector $\mathbf{v}$ is periodically exchanged over the network to synchronize the work between the different nodes. Now, assume in addition to the $K$ worker nodes we have $L$ GPUs available on each node. A naive way to parallelize the work would be to setup CoCoA with $KL$ workers, define $KL$ local subproblems and assign one subproblem to each GPU. This approach, however, would require the synchronization of $\mathbf{v}$ between all GPUs in every round of the algorithm and the performance would thus be limited be the slowest interconnect. To avoid this and take advantage of the fast interconnect amongst the GPUs of the same node, we propose a nested version of the CoCoA scheme which offers the possibility to perform multiple inner communication rounds within one outer communication round over the network. The local subproblems in the nested version are defined according to [18] where we recursively apply the separable approximation to the respective objectives. For an explicit statement of the local subproblems we refer the reader to the Appendix. To give convergence guarantees for our hierarchical scheme we refine the existing convergence results of CoCoA and combine these with tight convergence guarantees for the inner level of CoCoA derived by exploiting the specific structure of the subproblem objective.

Assume the local subtasks are solved $\theta$-approximately according to the definition introduced in [18]. Then, we can bound the suboptimality $\varepsilon := \mathcal{F}(\boldsymbol{\alpha}) - \min_{\boldsymbol{\alpha}} \mathcal{F}(\boldsymbol{\alpha})$ after $t_1$ iterations of CoCoA with

$t_2$ inner iterations each as

$$\mathbb{E}[\varepsilon] \leq \left[ \frac{4R^2 K \beta c_A}{1 - \left(1 - (1-\theta)\frac{1}{L}\right)^{t_2}} \right] \frac{1}{t_1} \tag{2}$$

where $\beta$ denotes the smoothness parameter of $f$, $R$ is a bound on the support of $g_i$ and $c_A := \|A\|^2$. For strongly convex $g_i$ a linear rate can be derived where we refer the reader to the appendix for detailed proofs. For the special case where we choose to only do a single update in the inner level, i.e., $t_2 = 1$, we recover the classical CoCoA scheme with $KL$ workers.

The benefit of the proposed hierarchical scheme becomes more significant if the discrepancy between the costs of intra-node and inter-node communication is large such as often found in modern cloud infrastructures. Let us assume there is a cost $c_1$ associated with communicating the shared vector $\mathbf{v}$ over the network and a cost $c_2$ with communicating $\mathbf{v}$ between the GPUs within a node. Then, for a given cost budget $C$, the right-hand side of (2) can be optimized for $t_1, t_2$ to achieve the best accuracy under a cost constraint $C \leq t_1 t_2 c_{\text{comp}} + t_1 c_1 + t_1 t_2 c_2$ where $c_{\text{comp}}$ denotes the cost of computing a $\theta$-approximate solution on the subtasks.

## 3   Implementation Details

In this section we will describe implementation details of Snap ML starting with details of the GPU-based local solver and working up to the high-level APIs. We have attempted to highlight the components that are most critical in terms of performance, in the hope that some of these ideas may be applicable to other machine learning software, including popular deep learning frameworks.

### 3.1   GPU Local Solver

To efficiently solve the optimization problem assigned to the GPU accelerators we implement the twice-parallel asynchronous stochastic coordinate descent solver (TPA-SCD) [15, 14].

*Extension for Logistic Regression.* In the previous literature [15, 14], TPA-SCD has been applied to ridge regression, lasso and support vector machines. These objectives have the desirable property that coordinate descent updates have closed-form solutions. In Snap ML, we also support objective functions for which this is not the case such as logistic regression. To address this issue, instead of solving the coordinate-wise subproblem exactly, we make a single step of Newton's method, using the previous value of the model as the initial point [22]. We find that the computations required to compute the Newton step (i.e, the first and second derivative) can also be expressed in terms of a simple inner product and thus the same TPA-SCD machinery can be applied.

*Adaptive Damping.* A challenge arises when applying the asynchronous TPA-SCD algorithm to dense datasets (or datasets which are globally sparse but locally dense in a few features) due to the fact that a thread block on the GPU may have an inconsistent view of the shared vector $\mathbf{v}$ if it is reading while another thread block has only partially written its updates to the same vector in memory. These inconsistencies can lead to divergence in the coordinate descent algorithm. To alleviate this issue we have implemented a damping heuristic, similar to that proposed in [23], to artificially slows down the model updates leading to more robust convergence behavior. We initialize the algorithm with the damping parameter set to 1 (i.e., no damping) and after every sub-epoch on the GPU, verify that the value of the local subproblem has actually decreased. If it has not, we discard the current round of model updates, halve the value of the damping parameter and proceed. We note that the damping parameter may be adapted differently across data partitions. This adaptive scheme introduces the cost needed to evaluate the value of the local subproblem within every sub-epoch, however this cost can be mostly amortized into the TPA-SCD kernel and only requires an additional reduce operation, for which we use the DeviceReduce operator provided by the CUB library [17].

### 3.2   Pipelining

*Asynchronous Data Streaming.* When the data partition of each node is too large to fit into the aggregate GPU memory on that node, we must employ out-of-core techniques to move the data in and out of GPU memory. One option is to split the data into batches and sequentially process each batch on the local GPUs. Snap ML also provides the ability to use DuHL [7] to dynamically

**CPU** | **INTERCONNECT** | **GPU**

Copy RN (i)

RNG for chunk (i+1) | Copy data chunk (i+1) | Sort RN (i) by key / TPA-SCD on chunk (i)

Copy RN (i+1)

RNG for chunk (i+2) | Copy data chunk (i+2) | Sort RN (i+1) by key / TPA-SCD on chunk (i+1)

Copy RN (i+2)

RNG for chunk (i+3) | Copy data chunk (i+3) | Sort RN (i+2) by key / TPA-SCD on chunk (i+2)

Figure 2: Data streaming pipeline.

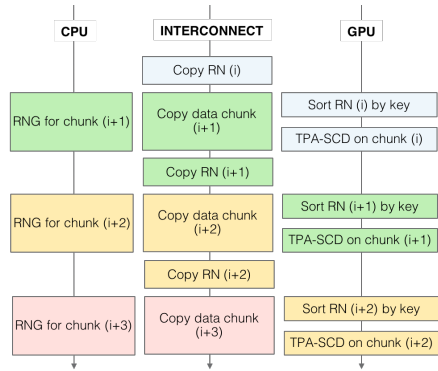

Figure 3: Example of snap-ml-mpi API.

determine which set of data points that are most beneficial to move into the GPU memory as the training progresses. Both of these schemes involve moving data over the CPU-GPU interconnect and while, for sparse models, it has been shown that when using DuHL the amount of data that needs to be copied reduces with the number of rounds, there can still be some significant overheads related to data transfer for dense models. To alleviate these overheads we have developed an alternative, more hardware-optimized approach. We partition the GPU memory into an active buffer and a swap buffer. Then, using CUDA streams, we can perform TPA-SCD on the data in the active buffer while at the same time copying the next batch of data into the swap buffer. As we shall show in Section 4, this pipelined approach can allow one to completely hide the data transfer time behind the computation time when using high-speed interconnects such as NVLINK.

*Streaming Permutation Generation.* In order to implement TPA-SCD, we must generate a permutation of the coordinates in the active buffer. In order to fully utilize the available hardware, we introduce a third pipeline stage to the training algorithm whereby the CPU is used to generate a set of 32-bit pseudo-random numbers for the batch of data that is currently being transferred into the swap buffer. At the start of the next round, we copy the random numbers onto the GPU device and use the DeviceRadixSort operator provided by CUB to sort the integers by index thus resulting in the required permutation in GPU memory. The sorting function templates provided by CUB have a significant advantage over those provided by the Thrust library [12] in that they properly support CUDA streams. Thrust's sort by key internally allocates memory which is a blocking operation on the GPU, whereas CUB explicitly requires that all memory be allocated upfront. The resulting 3-stage pipeline is illustrated in Figure 2. In order to ensure that the pseudo-random number generator does not become a bottleneck we implement a multi-threaded version of the highly efficient XORSHIFT algorithm [9].

### 3.3 Communication Patterns

*Intra-node Communication.* Within a single node, communication of the shared vector between the GPUs is handled within a single process using multi-threading. A thread is spawned to manage each GPU and within each local iteration, the updated shared vector is copied onto all devices using asynchronous CUDA memcpy operations. The GPU performs its updates and the changes to the shared vector are asynchronously copied back to the CPU and aggregated. After a number of local iterations are completed, the local changes to the shared vector are aggregated over the network interface. How exactly the updates to the shared vector are aggregated depends on whether the Spark or MPI API to Snap ML is used, as described in the next section.

*Inter-node Communication.* When using Spark, each node is managed by a Spark executor and the changes to the global shared vector are copied over the JNI from the underlying shared library into the JVM memory space and aggregated using Spark's reduce operator. The data is represented using raw byte arrays in order to minimize serialization/deserialization cost. The updated value of the shared vector is then communicated to each node using Spark's broadcast operator. When using MPI, an MPI process is spawned on each node and the global shared vector is updated in place using MPI's Allreduce operator.

*NUMA Locality.* In order to achieve the maximal bandwidth provided by the CPU-GPU interconnect it is essential that the software framework is implemented in a NUMA-aware manner. Specifically, if

the software is deployed on a two-socket node in which two GPUs are attached to each socket, it is critical that the threads that manage data transfer to those GPUs be pinned to the correct socket. This can be easily enforced using the functionality provided in the MPI rankfile that assigns the cores to each MPI process.

### 3.4 Software Architecture

*C++ Template Library.* The core functionality of Snap ML is implemented in C++/CUDA as a header-only template library: *libglm*. It provides class templates for CPU, GPU and multi-GPU local solvers that can be instantiated with arbitrary data formats (e.g. sparse, dense, compressed) and arbitrary objective functions mapping (1).

*Local API.* We provide a Python module, *snap-ml-local*, that adheres to the scikit-learn API and can be used to accelerate training of GLMs in a non-distributed setting. This API is targeted at single-node users who wish to accelerate existing scikit-learn-based applications using one or more GPUs that are attached locally. This module exploits the functionality offered by libglm while being flexible in that it can be readily combined with additional functionality from scikit-learn such as data loading, pre-processing and evaluation metrics.

*MPI API.* For users with larger data, who wish to perform training in a distributed environment we provide an additional Python module: *snap-ml-mpi*. By importing this module, the users can describe their application using high-level Python code and then submit an MPI job on their cluster using mpirun specifying the nodes to be used for training. At run-time, the Python code makes calls to *libglm* via an intermediate C++ layer that executes MPI operations to coordinate the training. The module also provides functions for efficient distributed data loading and evaluation of performance metrics. An illustrative example is given in Figure 3.

*Spark API.* Finally, for users who wish to perform distributed training on Apache Spark-managed infrastructure we provide *snap-ml-spark*. This module is essentially a lightweight Py4J [5] wrapper around an underlying jar package that interacts with libglm via the Java Native Interface. Local data partitions are managed by *libglm* and reside in memory outside of the JVM, thus enabling efficient GPU acceleration. Apache Spark introduces a number of additional layers into the software stack and thus a number of associated overheads. For this reason, we typically observe that the performance of the Spark-based deployments of Snap ML are slower than those using MPI [6].

## 4 Experimental Results

In the following we will evaluate the performance of Snap ML and compare with some widely-used ML frameworks in a single-node environment and a multi-node environment. Additionally, we will provide an in-depth analysis of two key aspects: pipelining and hierarchical training.

*Application and Datasets.* We will focus on the application of click-through rate prediction (CTR), which is a binary classification task. For our multi-node experiments we will use the *Terabyte Click Logs* dataset (criteo-tb) released by Criteo Labs [3]. It consists of 4.2 billion examples with 1 million feature values. We use the data collected during the first 23 days for the training of our models and the last day for testing. The training data is 2.3TB in SVM Light format and is thus one of the largest publicly available datasets, making it ideal for evaluating the performance of distributed ML frameworks. For our single-node experiments we use the smaller dataset released by Criteo Labs as part of their 2014 Kaggle competition (criteo-kaggle); the dataset has 45 million training examples and 1 million features. We perform a random 75%/25% train/test split. We obtained the preprocessed data for both datasets from [2].

*Infrastructure.* The results in this section were obtained using a cluster of 4 IBM Power Systems* AC922 servers. Each server has 4 NVIDIA Tesla V100 GPUs attached via the NVLINK 2.0 interface. The nodes are connected via both an InfiniBand network as well as a slower 1Gbit Ethernet interface. For evaluation of the pipeline performance we also used an Intel x86-based machine (Xeon** Gold 6150 CPU@2.70GHz ) with a single NVIDIA Tesla V100 GPU attached using the PCI Gen3 interface.

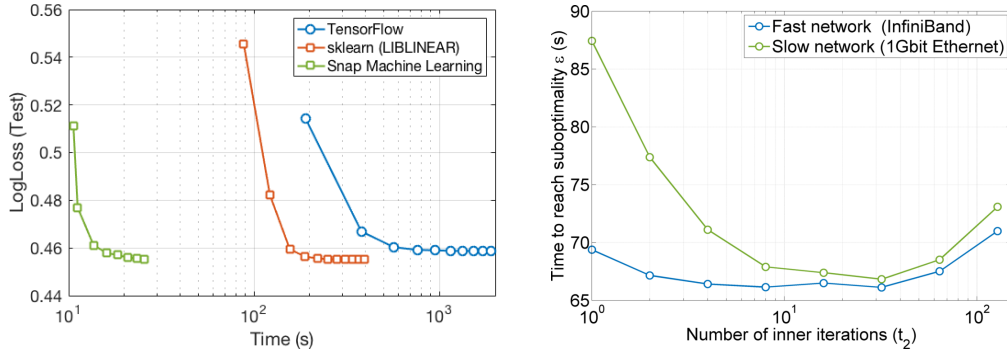

Figure 4: Single node benchmark (criteo-kaggle).   Figure 5: Performance of hierachical CoCoA.

## 4.1   Single-Node Performance

We benchmark the single node performance of Snap ML for the training of a logistic regression classifier against an equivalent solution in scikit-learn and TensorFlow. The same value of the regularization parameter was used in all cases. The different frameworks are used as follows:

*scikit-learn.* We load a pickled version of the dataset and train a logistic regression classifier in scikit-learn, with the option to solve the dual formulation enabled which allows faster training for this application. Under the hood, scikit-learn is calling the LIBLINEAR library [8] to solve the resulting optimization problem. It operates in single-threaded mode and can not leverage any available GPU resources.

*TensorFlow.* For the TensorFlow experiment, we first convert the svmlight data into the native binary format for TensorFlow (TFRecord) using a custom parser. We then feed the TFRecord to a TensorFlow binary classifier (tf.contrib.learn.LinearClassifier), treating the TFRecord features as sparse columns with integerized features. We use the stochastic dual coordinated ascent optimizer provided by TensorFlow, using the optimizer and train input function options suggested by Google [11]. We use a batch size of 1M, and a number of IO threads equal to the number of physical processors – settings which we have experimentally found to perform the best. The implementation is multi-threaded and can leverage GPU resources (for the classifier training and evaluation) if available. In this case we let TensorFlow use a single V100 GPU since we found it was faster than using all four.

*Snap ML.* We load a pickled version of the dataset and train a logistic regression classifier in Snap ML using the snap-ml-local API. To compare with TensorFlow, we only allow Snap ML to use a single GPU. Since the criteo-kaggle dataset fits into GPU memory, the streaming functionality of Snap ML is not active in this experiment.

In Figure 4, we compare the performance of the three aforementioned solutions. TensorFlow converges in approximately 500 seconds whereas scikit-learn takes around 200 seconds. This difference may be explained by the highly optimized C++ backend of scikit-learn for workloads that fit in memory, whereas TensorFlow processes data in batches [2]. Finally, we can see that Snap ML converges in around 20 seconds, an order of magnitude faster than the other frameworks.

## 4.2   Out-of-core Performance

In order to evaluate the streaming performance of Snap ML we train a logistic regression model using a single GPU for the first 200 million training examples of the criteo-tb dataset. We profile the execution on a machine that uses the PCI Gen 3 interconnect and a machine that uses the NVLINK 2.0 interconnect. In Figure 6a, we show the profiling results for the PCI-based setup. On stream S1, the random numbers for the next batch are copied (Init) and then the sorting and TPA-SCD are performed (Train chunk) - this takes around 90ms. In stream S2 we copy the next data chunk onto the GPU which takes around 318ms and is thus the bottleneck. In Figure 6b, for the NVLINK-based setup we observe that the copy time is reduced to 55ms (almost a factor of 6), due to the faster bandwidth provided by NVLINK 2.0. This speed-up hides the data copy time behind the kernel execution, effectively removing the copy time from the critical path and resulting in a 3.5x speed-up.

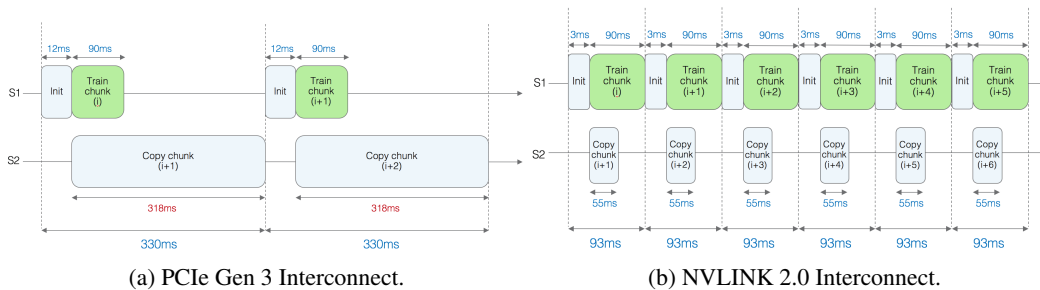

(a) PCIe Gen 3 Interconnect.            (b) NVLINK 2.0 Interconnect.

Figure 6: Pipelined performance 'out-of-core'.

## 4.3 Hierarchical Scheme

To evaluate the effect of the hierarchical application of CoCoA, we train the first billion examples of the criteo-tb dataset using all 16 GPUs of the cluster. We train a logistic regression model from snap-ml-mpi and evaluate the time to reach a target training suboptimality $\varepsilon$ as a function of the number of inner CoCoA iterations performed ($t_2$) using both a fast network (InfiniBand) and a slow network (1Gbit Ethernet). The scheme where $t_2 = 1$ corresponds to the standard non-hierarchical CoCoA approach. The results, as plotted in Figure 5, show that there is only little benefit to setting $t_2 > 1$ when using the fast network since communication cost only accounts for a small fraction of the overall training time, but when using the slow network it is possible to approach the fast network performance by increasing $t_2$. Such a scheme is therefore suitable for use in cloud-based deployments where high-performance networking is not normally available.

## 4.4 Tera-Scale Benchmark

To evaluate the performance of Snap ML on criteo-tb, we use the snap-ml-mpi interface to train a logistic regression classifier using all 16 GPUs in the cluster. Because the data does not fit into the aggregated memory of the GPUs the streaming functionality of Snap ML are active in this experiment. We obtain a logarithmic loss on the test set of 0.1292 in 1.53 minutes. This is the total runtime including data loading, initialization and training time.

There have been a number of previously-published results on this same benchmark, using different machine learning software frameworks, as well as different hardware resources. We will briefly review these results:

- *LIBLINEAR.* In an experimental log posted in the libsvm datasets repository [2], the authors report using LIBLINEAR-CDBLOCK [21] to perform training on a single machine with 128GB of RAM.

- *Vowpal Wabbit.* In [19], the authors evaluated the performance of Vowpal Wabbit, a fast out-of-core learning system. Training was performed on a 12 core (24 thread) machine with 128GB of memory using Vowpal Wabbit 8.3.0 using the first 3 billion training examples of criteo-tb.

- *Spark MLlib.* In the same benchmark [19], the authors also measured the performance of the logistic regression provided by Spark MLlib. They deploy Spark 2.1.0 a cluster with total 512 cores and 2TB of memory. Each executor is giving 4 cores and 16TB of memory.

- *TensorFlow.* Google have also published results where they use Google Cloud Platform to scale out the training of a logistic regression classifier from TensorFlow [20]. They report using 60 workers machines and 29 parameter machines for the training of the full dataset.

- *TensorFlow on Spark.* Criteo have published code [4] to train a logistic regression model that uses Tensorflow together with Spark for distributing the training across multiples node. They also provide results that were obtained using 12 Spark executors.

In Figure 7, we provide a visual summary of these results. We can observe that Snap ML on 16 GPUs is capable of training such a model to a similar level of accuracy, 46x faster than the best previously reported results, which was obtained using TensorFlow. In addition to the previously

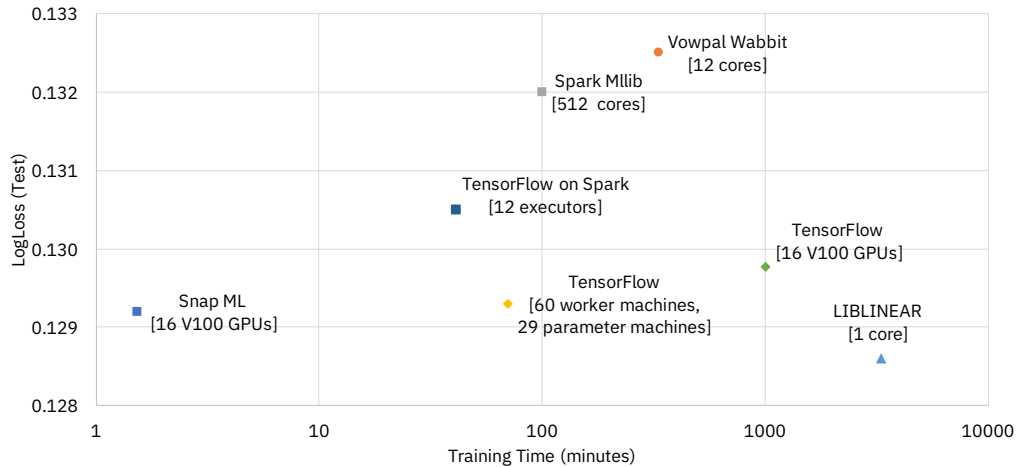

Figure 7: Previously-published results for logistic regression on the Terabyte Click Logs dataset.

published results, we have also reproduced the TensorFlow results on our infrastructure, using the optimizer and train input function options suggested by Google [11] (similar to 4.1). We tuned the batch size used by TensorFlow and found that Snap ML can train the logistic regression classifier over $500\times$ faster than TensoFlow on exactly the same hardware.

## 5    Conclusions

In this work we have described Snap ML, a new framework for fast training of generalized linear models. Snap ML can exploit modern computing infrastructure consisting of multiple machines that contain both CPUs and GPUs. The framework is hierarchical in nature, allowing it to adapt to cloud-based deployments where the cost of communication between nodes may be relatively high. It is also able to effectively leverage modern high-speed interconnects to hide the cost of transferring data between CPU and GPU when training on datasets that are too large to fit into GPU memory. We have shown that Snap ML can provide significantly faster training than existing frameworks in both single-node and multi-node benchmarks. On one of the largest publicly available datasets, we have shown that Snap ML can be used to train a logistic regression classifier in 1.5 minutes: more than an order of magnitude faster than any of the previously reported results.

## Acknowledgement

The authors would like to thank Martin Jaggi for valuable input regarding the algorithmic structure of our system, Michael Kaufmann and Adrian Schüpbach for testing and bug fixes, Kubilay Atasu for contributing code for load balancing, and Manolis Sifalakis and Urs Egger for setting up vital infrastructure. We would also like to thank our colleagues Christoph Hagleitner and Cristiano Malossi for providing access to heterogeneous compute resources and providing valuable support when scheduling large-scale jobs. Finally, we would also like to thank Hillery Hunter, Paul Crumley and I-Hsin Chung for providing access to the servers that were used to perform the tera-scale benchmarking, and Bill Armstrong for his guidance and support of this project.

## Footnotes

[2]We did try to load the whole dataset in TensorFlow and not use batching, but there seems to be a known issue with TensorFlow for datasets that are bigger than 2GB [13].

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
