[Supplementary Material]

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

# A Convergence Analysis

In this section we derive the convergence result presented in (2) for the hierarchical CoCoA scheme proposed in this paper. Therefore we first give some background on the classical CoCoA scheme, its convergence guarantees, and then derive a new convergence rate for our hierarchical method.

## A.1 CoCoA

The CoCoA framework has been proposed in [18]. It is designed to solve generalized linear models of the form (1) distributedly across $K$ worker nodes where each worker has access to its local partition of the training data. In particular, each worker has access to a subset $\mathcal{I}_k$ of the columns of the data matrix $A$ where $\mathcal{I}_k$ are disjoint index sets such that

$$\bigcup_k \mathcal{I}_k = [n], \ \ \mathcal{I}_i \cap \mathcal{I}_j = \emptyset \ \ \forall i \neq j$$

and $n_k := |\mathcal{I}_k|$ denotes the size of partition $k$. Hence, each machine stores in its memory the submatrix $A^{(k)} \in \mathbb{R}^{d \times n_k}$ corresponding to its partition $\mathcal{I}_k$.

The CoCoA algorithm is an iterative procedure where in every round an update $\Delta\boldsymbol{\alpha}$ to the model is computed in a distributed manner. The computation of $\Delta\boldsymbol{\alpha}$ is split across the $K$ workers where each worker computes an update to its dedicated set of cordinates $\mathcal{I}_k$. To compensate for possible correlations between partitions and guarantee convergence the parameter $\sigma$ is introduced

$$\sigma \geq \sigma_{\max} := \max_{\mathbf{x} \in \mathbb{R}^n} \frac{\|A\mathbf{x}\|^2}{\sum_k \|A\mathbf{x}_{[k]}\|^2}$$

where we use $\mathbf{x}_{[k]}$ to denote a vector $\mathbf{x}$ with non-zero elements only for $i \in \mathcal{I}_k$.

**Local Subproblems** Worker $k \in [K]$ is assigned the following local subproblem:

$$\underset{\Delta\boldsymbol{\alpha}_{[k]}}{\arg\min} \ \mathcal{F}_\sigma^{(k)}(\Delta\boldsymbol{\alpha}_{[k]}; \boldsymbol{\alpha}, \mathbf{v}) \tag{3}$$

with

$$\mathcal{F}_\sigma^{(k)}(\Delta\boldsymbol{\alpha}_{[k]}; \boldsymbol{\alpha}, \mathbf{v}) := \frac{1}{K} f(\mathbf{v}) + \nabla f(\mathbf{v}) A\Delta\boldsymbol{\alpha}_{[k]} + \frac{\sigma\beta}{2} \|A\Delta\boldsymbol{\alpha}_{[k]}\|^2 + \sum_{i \in \mathcal{I}_k} g_i((\boldsymbol{\alpha} + \Delta\boldsymbol{\alpha}_{[k]})_i). \tag{4}$$

where $\beta$ denotes the smoothness parameter of $f$. The local subproblems (4) are independent for every $k \in [K]$ and can thus be solved in parallel. Furthermore, solving (3) only requires access to the local partition of the training data $A_{[k]}$ and the vector $\mathbf{v} := A\boldsymbol{\alpha}$ which is updated and shared among all workers during the algorithm. To analyze the convergence of our scheme we will use the following notion:

**Definition 1** ($\theta$-approximate [18])**.** For some $\theta \in [0, 1)$ the update $\Delta\boldsymbol{\alpha}_{[k]}$ is a $\theta$-approximate solution to the local suproblem (3) iff

$$\mathcal{F}_\sigma^{(k)}(\Delta\boldsymbol{\alpha}_{[k]}; \boldsymbol{\alpha}, \mathbf{v}) - \mathcal{F}_\sigma^{(k)}(\Delta\boldsymbol{\alpha}_{[k]}^\star; \boldsymbol{\alpha}, \mathbf{v}) \leq \theta \left[ \mathcal{F}_\sigma^{(k)}(\mathbf{0}; \boldsymbol{\alpha}, \mathbf{v}) - \mathcal{F}_\sigma^{(k)}(\Delta\boldsymbol{\alpha}_{[k]}^\star; \boldsymbol{\alpha}, \mathbf{v}) \right]$$

where $\Delta\boldsymbol{\alpha}_{[k]}^\star = \arg\min_{\Delta\boldsymbol{\alpha}_{[k]}} \mathcal{F}_\sigma^{(k)}(\Delta\boldsymbol{\alpha}_{[k]}; \boldsymbol{\alpha}, \mathbf{v})$

---

**Algorithm 1** CoCoA [18]

---

1: **Input:** Data matrix $A$ distributed column-wise according to partition $\{\mathcal{I}_k\}_{k=1}^K$. parameter $\sigma$ for the local subproblems $\mathcal{F}_\sigma^{(k)}$.
   Starting point $\boldsymbol{\alpha}^{(0)} := \mathbf{0} \in \mathbb{R}^n$, $\mathbf{v}^{(0)} := \mathbf{0} \in \mathbb{R}^d$.
2: **for** $t = 0, 1, 2, \ldots$ **do**
3:   **for** $k \in \{1, 2, \ldots, K\}$ **in parallel over workers do**
4:     $\Delta\boldsymbol{\alpha}_{[k]} \leftarrow \theta$-approximate solution to the local subproblem (3).
5:     update local variables $\boldsymbol{\alpha}_{[k]}^{(t+1)} := \boldsymbol{\alpha}_{[k]}^{(t)} + \Delta\boldsymbol{\alpha}_{[k]}$
6:     return updates to shared state $\Delta\mathbf{v}_k := A\Delta\boldsymbol{\alpha}_{[k]}$
7:   **end for**
8:   reduce $\mathbf{v}^{(t+1)} := \mathbf{v}^{(t)} + \sum_{k=1}^K \Delta\mathbf{v}_k$
9: **end for**

---

## A.2 Convergence Guarantees

The following two theorems define the convergence behavior of CoCoA for strongly convex and non-strongly convex $g_i$. We define the suboptimality of (1) as $\varepsilon^{(t)} := \mathcal{F}(\boldsymbol{\alpha}^{(t)}) - \mathcal{F}(\boldsymbol{\alpha}^\star)$

**Theorem 1** (based on [18](Theorem 3)). *Consider the CoCoA Algorithm as defined in Algorithm 1. Let $\theta$ be the approximation quality of the local solver according to Definition 1. Let $f$ be $\beta$-smooth and $g_i$ $\mu$-strongly convex. Then the suboptimality after $t$ iterations can be bounded as*

$$\varepsilon^{(t)} \leq \left(1 - (1-\theta)\frac{\mu}{\mu + \sigma\beta c_A}\right)^t \varepsilon^{(0)}.$$

**Theorem 2** (based on [18](Theorem 2)). *Consider the CoCoA Algorithm as defined in Algorithm 1. Let $\theta$ be the approximation quality of the local solver according to Definition 1. Let $f$ be $\beta$-smooth and $g_i$ be a convex function with $R$-bounded support. Then the suboptimality after $t \geq 1$ iterations can be bounded as*

$$\mathbb{E}\big[\varepsilon^{(t)}\big] \leq \frac{4R^2 c_A \sigma\beta}{(1-\theta)}\frac{1}{t}$$

*Proof.* For the proof of Theorem 1 see [18]. For the proof Theorem 2 we follow the proof of Theorem 9 in [18] but when choosing the free parameter $s$ we use the explicit minimizer which simplifies the final rate. □

## B Hierarchical CoCoA

We introduce a second level of CoCoA, where the local subproblems (3) are solved distributedly across $L$ compute units, e.g., across multiple GPUs within one node. Let us without loss of generality focus on a particular worker $k \in [K]$ and for reasons of readability we denote the local data partition as $B := A_{[k]} \in \mathbb{R}^{d \times n_k}$. Then, with a change of variables the local subproblem (4) can be rewritten as

$$\arg\min_{\mathbf{d}} \mathcal{G}(\mathbf{d}; \boldsymbol{\alpha}, \mathbf{v}) := \frac{1}{K}f(\mathbf{v}) + \nabla f(\mathbf{v})^\top B\mathbf{d} + \frac{\sigma\beta}{2}\|B\mathbf{d}\|^2 + \sum_{i \in \mathcal{I}_k} g_i(\alpha_i + d_{j(i)}) \tag{5}$$

where $j(.)$ enumerates the elements in $\mathcal{I}_k$. In order to apply a nested CoCoA to this problem we map (5) to the general framework (1). Therefore we choose $\bar{f}(B\mathbf{d}) := \frac{1}{K}f(\mathbf{v}) + \nabla f(\mathbf{v})^\top B\mathbf{d} + \frac{\sigma\beta}{2}\|B\mathbf{d}\|^2$, $\bar{g}_i(d_i) = g_i(\alpha_{\mathcal{I}_{k\,i}} + d_i)$ such that our local optimization task becomes

$$\arg\min_{\mathbf{d}} \bar{f}(B\mathbf{d}) + \sum_i \bar{g}_i(d_i).$$

Note that $\bar{f}$ is $\bar{\beta} = \beta\sigma$-smooth and $\bar{g}_i$ is $\bar{\mu} = \mu$-strongly convex. We introduce the separability parameter $\bar{\sigma}$ on the local data partition, i.e.,

$$\bar{\sigma} \geq \bar{\sigma}_{\max} := \max_{\mathbf{x} \in \mathbb{R}^{n_k}} \frac{\|A_{[k]}\mathbf{x}\|^2}{\sum_\ell \|A_{[k]}\mathbf{x}_{[\ell]}\|^2}.$$

Thus we can define local subtasks according to (3): Let $\bar{\mathbf{v}} := B\mathbf{d}$ be the local shared vector and the local subtasks are defined as

$$\arg\min_{\Delta\mathbf{d}_{[\ell]}} \mathcal{G}_{\bar{\sigma}}^{(\ell)}(\Delta\mathbf{d}_{[\ell]}; \mathbf{d}, \bar{\mathbf{v}})$$

where

$$\begin{aligned}
\mathcal{G}_{\bar{\sigma}}^{(\ell)}(\Delta\mathbf{d}_{[\ell]}; \mathbf{d}, \bar{\mathbf{v}}) &:= \frac{1}{L}\bar{f}(\bar{\mathbf{v}}) + \nabla\bar{f}(\bar{\mathbf{v}})^\top B\Delta\mathbf{d}_{[\ell]} + \frac{\bar{\beta}\bar{\sigma}}{2}\|B\Delta\mathbf{d}_{[\ell]}\|^2 + \sum_{i \in \mathcal{I}_{k,\ell}} \bar{g}_i((\mathbf{d}_{[\ell]} + \Delta\mathbf{d}_{[\ell]})_i) \\
&= \frac{1}{L}\left[\frac{1}{K}f(\mathbf{v}) + \nabla f(\mathbf{v})\bar{\mathbf{v}} + \frac{\beta\sigma}{2}\|\bar{\mathbf{v}}\|^2\right] + \left[\nabla f(\mathbf{v}) + \beta\sigma\bar{\mathbf{v}}\right]^\top B\Delta\mathbf{d}_{[\ell]} \\
&\quad + \frac{\bar{\sigma}\bar{\beta}}{2}\|B\Delta\mathbf{d}_{[\ell]}\|^2 + \sum_{i \in \mathcal{I}_{k,\ell}} \bar{g}_i((\mathbf{d}_{[\ell]} + \Delta\mathbf{d}_{[\ell]})_i)
\end{aligned} \tag{6}$$

## B.1 Convergence

Let us denote $\bar{\varepsilon}$ the suboptimality of the local subproblem (3), i.e., $\bar{\varepsilon} := \mathcal{G}(\mathbf{d}; \boldsymbol{\alpha}, \mathbf{v}) - \mathcal{G}^\star$ where $\mathcal{G}^\star = \min_{\mathbf{d}} \mathcal{G}(\mathbf{d}; \boldsymbol{\alpha}, \mathbf{v})$. Assume the subtasks (6) are solved $\bar{\theta}$-approximately $\forall \ell$. Then, we can prove the following convergence guarantee for the inner level of CoCoA, i.e., CoCoA running on (3):

$$\bar{\varepsilon} \leq \left(1 - (1 - \bar{\theta}) \frac{\beta \sigma c_A + \mu}{\bar{\sigma} \bar{\beta} c_A + \mu}\right)^t \bar{\varepsilon}^{(0)} \tag{7}$$

**Remark 1.** *Note that this rate improves over the classical CoCoA rate of Theorem 1 since it exploits the quadratic structure of the objective given by the local subproblem (5).*

*Proof.* We first recall that by the definition of the CoCoA local subtasks they upper bound the objective as follows:

$$\mathcal{G}(\mathbf{d} + \Delta \mathbf{d}) \leq \sum_\ell \mathcal{G}_{\bar{\sigma}}^{(\ell)}(\Delta \mathbf{d}_{[\ell]}; \mathbf{d}) \tag{8}$$

Note that we dropped the implicit dependence of the objectives $\mathcal{G}$ and $\mathcal{G}_{\bar{\sigma}}^{(\ell)}$ on $\mathbf{v}, \boldsymbol{\alpha}$ for reasons of readability. Now we exploit that the individual subtasks $\mathcal{G}_{\bar{\sigma}}^{(\ell)}$ are solved $\bar{\theta}$-approximately which yields

$$
\begin{aligned}
\mathcal{G}(\mathbf{d} + \Delta \mathbf{d}) - \mathcal{G}^\star &\leq \sum_\ell \mathcal{G}_{\bar{\sigma}}^{(\ell)}(\Delta \mathbf{d}_{[\ell]}; \mathbf{d}) - \mathcal{G}^\star \\
&= \mathcal{G}(\mathbf{d}) - \mathcal{G}^\star - \left(\mathcal{G}(\mathbf{d}) - \sum_\ell \mathcal{G}_{\bar{\sigma}}^{(\ell)}(\Delta \mathbf{d}_{[\ell]}; \mathbf{d})\right) \\
&\leq \mathcal{G}(\mathbf{d}) - \mathcal{G}^\star - (1 - \bar{\theta}) \underbrace{\left(\mathcal{G}(\mathbf{d}) - \min_{\Delta \mathbf{d}_{[\ell]}} \sum_\ell \mathcal{G}_{\bar{\sigma}}^{(\ell)}(\Delta \mathbf{d}_{[\ell]}; \mathbf{d})\right)}_{\Lambda}
\end{aligned} \tag{9}
$$

where $\mathcal{G}^\star := \min_{\Delta \mathbf{d}} \mathcal{G}(\Delta \mathbf{d}; \mathbf{d})$. Plugging in the definitions of $\mathcal{G}$ and $\mathcal{G}_{\bar{\sigma}}^{(\ell)}$ we find

$$\Lambda := \min_{\Delta \mathbf{d}} \nabla f(\mathbf{v})^\top B \Delta \mathbf{d} + \beta \sigma \bar{\mathbf{v}}^\top B \Delta \mathbf{d} + \frac{\bar{\sigma} \bar{\beta}}{2} \sum_\ell \|B \Delta \mathbf{d}_{[\ell]}\|^2 + \sum_{i \in \mathcal{I}_{k,\ell}} \bar{g}_i((\mathbf{d}_{[\ell]} + \Delta \mathbf{d}_{[\ell]})_i) - \bar{g}_i((\mathbf{d}_{[\ell]})_i)$$

We proceed by bounding the term $\Lambda$. Therefore we consider a not necessarily optimal update $\Delta \mathbf{d} = \lambda(\mathbf{x} - \mathbf{d})$ where $\mathbf{x} \in \mathbb{R}^{n_k}$ will be specified favorably in the course of the proof. Thus, the following inequality holds for every $\lambda \in (0, 1]$ and every $\mathbf{x} \in \mathbb{R}^{n_k}$:

$$
\begin{aligned}
\Lambda &\leq \min_\lambda \lambda \nabla f(\mathbf{v})^\top B(\mathbf{x} - \mathbf{d}) + \lambda \beta \sigma \bar{\mathbf{v}}^\top B(\mathbf{x} - \mathbf{d}) + \frac{\bar{\sigma} \bar{\beta} \lambda^2}{2} \sum_k \|B(\mathbf{x} - \mathbf{d})_{[k]}\|^2 \\
&\quad + \sum_i \bar{g}_i((1 - \lambda)d_i + \lambda x_i) - \bar{g}_i(d_i)
\end{aligned} \tag{10}
$$

Now using $\mu$-strong-convexity of $g_i$ we have

$$\sum_i \bar{g}_i((1 - \lambda)d_i + \lambda x_i) - \bar{g}_i(d_i) \leq \sum_i -\lambda \bar{g}_i(d_i) + \lambda \bar{g}_i(x_i) - \frac{\bar{\mu} \lambda(1 - \lambda)}{2} \|\mathbf{x} - \mathbf{d}\|^2$$

Further augmenting the first term in (10) to extract the subproblem objective we find

$$
\begin{aligned}
\Lambda &\leq \min_\lambda \lambda \left[\mathcal{G}(\mathbf{x}) - \mathcal{G}(\mathbf{d})\right] - \lambda \frac{\beta \sigma}{2} \|B\mathbf{x}\|^2 + \lambda \frac{\beta \sigma}{2} \|B\mathbf{d}\|^2 \\
&\quad + \lambda \beta \sigma \bar{\mathbf{v}}^\top B(\mathbf{x} - \mathbf{d}) + \frac{\bar{\sigma} \bar{\beta} \lambda^2}{2} \sum_k \|B(\mathbf{x} - \mathbf{d})_k\|^2 - \frac{\bar{\mu} \lambda(1 - \lambda)}{2} \|\mathbf{x} - \mathbf{d}\|^2
\end{aligned}
$$

Now note that $\bar{\mathbf{v}} = B\mathbf{d}$ and completing the square we find

$$\Lambda \leq \min_\lambda \lambda \left(\mathcal{G}(\mathbf{x}) - \mathcal{G}(\mathbf{d})\right) + \left[\frac{\bar{\sigma} \bar{\beta} \lambda^2 c_A}{2} - \lambda \frac{\beta \sigma c_A}{2} - \frac{\bar{\mu} \lambda(1 - \lambda)}{2}\right] \|\mathbf{x} - \mathbf{d}\|^2$$

where we define $c_A$ such that $\|B\mathbf{d}\|^2 \leq c_A \|\mathbf{d}\|^2$. To finalize the proof we let $\mathbf{x} = \arg \min_{\mathbf{d}} \mathcal{G}(\mathbf{d})$, choose $\lambda = \frac{\beta \sigma c_A + \mu}{\bar{\sigma} \bar{\beta} c_A + \mu}$ and combe this with (9) which yields the following recursion:

$$\mathcal{G}(\mathbf{d} + \Delta \mathbf{d}) - \mathcal{G}^\star \leq \left(1 - (1 - \bar{\theta}) \frac{\beta \sigma c_A + \mu}{\bar{\sigma} \bar{\beta} c_A + \mu}\right)^{\bar{t}} (\mathcal{G}(\mathbf{0}) - \mathcal{G}^\star)$$

$\square$

**End-to-End Rate.** Recall the definition of $\theta$-approximate solutions from Definition 1. Let us denote the number of outer iterations by $t_1$ and the number of inner iterations between two outer iterations by $t_2$. Then, from (7) we know that after $t_2$ inner iterations the local subproblems (3) are solved with an accuracy

$$\theta = \left(1 - (1 - \bar{\theta})\frac{\beta\sigma c_A + \mu}{\bar{\sigma}\sigma\beta c_A + \mu}\right)^{t_2}.$$

Thus, combining this with Theorem 1 we can bound the suboptimality $\varepsilon$ after $t_1$ outer with $t_2$ inner iterations for strongly convex $g_i$ as

$$\varepsilon \quad \leq \quad \left(1 - \left[1 - \left(1 - (1 - \bar{\theta})\frac{\beta\sigma c_A + \mu}{\bar{\sigma}\sigma\beta c_A + \mu}\right)^{t_2}\right]\frac{\mu}{\sigma\beta c_A + \mu}\right)^{t_1}\varepsilon^{(0)}$$

and similarly we can bound the suboptimality for general non-strongly convex $g_i$ as

$$\mathbb{E}\big[\varepsilon\big] \leq \frac{4R^2 c_A \sigma\beta}{\left(1 - \left(1 - (1 - \bar{\theta})\frac{1}{\bar{\sigma}}\right)^{t_2}\right)}\frac{1}{t_1}$$

with $R$ such that $\|\boldsymbol{\alpha}\| \leq R$ for every iterate.

**Remark 2.** *To obtain the rate (2) we note that $\sigma_{max} \leq K$ and $\bar{\sigma}_{max} \leq L$.*

**Remark 3.** *For the special case where $t_2 = 1$ we can recover the rate of single-level CoCoA from Theorem 1 and 2 with $KL$ workers.*