[Reviews · NeurIPS 2018]

Reviewer 1



The work at hand describes a new distributed implementation (Snap ML) for training generalized linear models for very large datasets. More precisely, the authors extend the popular CoCoA method [15] by providing a hierarchical version that is optimized for distributed computing environments. One of the key ingredients of this new version is the reduction of the overhead caused by the induced inter-node communication (see Section 2.1). In addition to a theoretical analysis of their new approach (Equation (2)), the authors also provide various implementation details including details related to an efficient local GPU solver per compute node (Section 3.1), to buffering techniques to reduce the overhead caused by memory transfers between host and device per compute node (Section 3.2), to an efficient exchange of information between compute nodes (Section 3.3), and to the overall software architecture. The experimental evaluation indicates some benefits of the new hierarchical scheme (Figure 5). It also demonstrates the benefits of the new framework for the Terabyte Click Logs. The paper is well written and technically sound. I also like the theoretical analysis (Appendix B) of the inter-node communication pattern that is induced by parallelyzing the work per compute node (i.e., by having two loops of parallelism). It think the paper definitely meets the very high NIPS standards. My two main concerns are (1) lack of novelty and (2) the experimental evaluation: (1) The framework extends the CoCoA framework. While being technically and theoretically sound, I find the practical merits of the approach quite limited. In particular, I am not blown away by the performance gains obtained via the new hierarchical framework. For instance, even given a slow network, I do not find the runtime reduction significant (e.g., 87 seconds instead for t_2=1 compared to 67 seconds for t_2 \approx 10, see Figure 5). I have the feeling that a similar performance gain could be achieved by compressing a shared vector v prior to sending it over the network. (2) The approach is clearly of practical importance. However, I was not fully convinced by the experimental evaluation. For instance, in Figure 4, I find the performance comparison between Snap ML, Scikit-Learn, and TensorFlow not entirely fair: Scikit-Learn only makes use of a CPU instead of a V100 GPU. Further, TensorFlow reads the data in batches from disk (as stated by the authors). Thus, Snap ML is faster than Scikit-Learn simply due to the GPU and faster than TensorFlow due to the slow transfer between disk and memory device. Can one improve the performance of TensorFlow by increasing the batch size or by buffering/caching? Also, while I find the results in Figure 7 impressive, I am missing one important baseline: Snap ML with t_2=1, that is, a Spark+GPU version of the previous CoCoA framework (without the additional hierarchical additions). What is the performance of this variant? Do the modifications proposed in the work yield significant better results compared to a rather direct distributed application of the existing framework (i.e., a direct Spark/GPU implementation of the CoCoA implementation)? Based on the authors' comments, I have decided to increase my score a bit. Overall, it is a decent contribution. I nevertheless think that the experimental evaluation could be improved. Minor comments: - Figure 5: hierarchical

Reviewer 2



This paper describes Snap ML, a framework that improves training time for generalized linear models in a distributed computing setup comprising of CPUs and GPUs using large datasets. The contributions include communication-cost aware design that allows for significant gains in the training time compared to existing relevant frameworks. There are several strengths of this work: - Snap ML builds on multiple state-of-the art algorithmic building blocks that allows it to make progress leveraging existing advances. - The evaluation is thorough. The authors have used a large dataset to evaluate the work and have provided a comparison with other existing approaches. SnapML achieves significant gains in improving training time. - The paper is generally pretty readable, but could use some more finishing. Especially the description of convergence in section 2.1 did not seem entirely clear (this might be due to space-limitation). There are also some feasible area of improvement: - I suggest devoting some space to explicitly note the challenges in implementing Snap ML starting with the non-hierarchical version of CoCoA. - Since Snap ML enables cloud-hosted deployments for training of generalized linear models, a discussion on the inherent performance variability observed in public cloud environments (due to shared, commodity hardware, and lack of complete control over the underlying resource to the users) should help make a stronger case. - Some experiments in public cloud setting, such as AWS EC2, using VMs that offer GPUs should definitely make it more convincing. - A comparison with standalone manycore architecture should strengthen the paper.

Reviewer 3



Paper describes a new software library focused on fast training of generalized linear models. It describes details of the architecture, how it maps to modern computing systems (with GPUs) and what specific aspects of the design and implementation are critical in achieving high performance. Finally they evaluate the performance of their implementation against existing popular implementations and show how SnapML is much faster in achieving the same results. There are many libraries for various ML algorithms available today, and SnapML focuses on one class of algorithms (Generalized Linear Models) and pushes the envelope for these algorithms. Strengths: - Well written paper that articulates the details of what's important. - Combines a number of optimizations seen across different systems but put together here in a good way. - Leverage GPUs well for these problems - something that hasn't been done well before. - e.g. implements a custom kernel for TPA-SCD on GPUs, and extends it to Logistic Regression while adding Adaptive damping. - Strong results compared to the state of the art for this target. Weaknesses: - Some of the baselines seem to use prior runs, but aren't quite fair. - For single node performance - SnapML impl caches dataset on GPU - Scikitlearn caches everything in memory - TensorFlow impl reads from disk - it is quite easy to cache entire dataset in memory (and likely in GPU memory) for this too. - This makes the comparison all about I/O - will be good to get a better comparison. - For both these setups it is good to highlight which of these optimizations got each of the performance wins. While this has been done briefly for SnapML as part of their explanations, some of these optimizations e.g. the TPA-SCD kernel and NVLINK could be used with TensorFlow quite easily as well. Hence the win maybe less because of the "overall library", than the specifics that are being talked about here. Overall the description and implementation of the optimization for GLMs seems like a good step forward, and it seems worthwhile to accept this paper. However it will be good to improve some of the baselines for comparison - even if they are shown separately as - improved published benchmarks via X, Y and Z, so the authors get full credit for the new numbers.